# Neuroinflammation-Associated Alterations of the Brain as Potential Neural Biomarkers in Anxiety Disorders

**DOI:** 10.3390/ijms21186546

**Published:** 2020-09-07

**Authors:** Eunsoo Won, Yong-Ku Kim

**Affiliations:** 1Department of Psychiatry, CHA Bundang Medical Center, CHA University, Seongnam 13496, Korea; eunsooowon@gmail.com; 2Department of Psychiatry, Korea University Ansan Hospital, Korea University College of Medicine, Ansan 15355, Korea

**Keywords:** panic disorder, agoraphobia, generalized anxiety disorder, neuroinflammation, neural biomarker, anti-inflammatory interventions

## Abstract

Stress-induced changes in the immune system, which lead to neuroinflammation and consequent brain alterations, have been suggested as possible neurobiological substrates of anxiety disorders, with previous literature predominantly focusing on panic disorder, agoraphobia, and generalized anxiety disorder, among the anxiety disorders. Anxiety disorders have frequently been associated with chronic stress, with chronically stressful situations being reported to precipitate the onset of anxiety disorders. Also, chronic stress has been reported to lead to hypothalamic–pituitary–adrenal axis and autonomic nervous system disruption, which may in turn induce systemic proinflammatory conditions. Preliminary evidence suggests anxiety disorders are also associated with increased inflammation. Systemic inflammation can access the brain, and enhance pro-inflammatory cytokine levels that have been shown to precipitate direct and indirect neurotoxic effects. Prefrontal and limbic structures are widely reported to be influenced by neuroinflammatory conditions. In concordance with these findings, various imaging studies on panic disorder, agoraphobia, and generalized anxiety disorder have reported alterations in structure, function, and connectivity of prefrontal and limbic structures. Further research is needed on the use of inflammatory markers and brain imaging in the early diagnosis of anxiety disorders, along with the possible efficacy of anti-inflammatory interventions on the prevention and treatment of anxiety disorders.

## 1. Introduction

As a group, anxiety disorders are the most common class of disorders listed in the Diagnostic and Statistical Manual of Mental Disorders, Fifth Edition. They are also the most costly mental disorders [1]. According to the Diagnostic and Statistical Manual of Mental Disorders, Fifth Edition, anxiety disorders include panic disorder, agoraphobia, generalized anxiety disorder, social anxiety disorder, specific phobia, selective mutism, separation anxiety disorder, substance/medication-induced anxiety disorder, and anxiety disorder due to another medical condition [2]. Continuous research has been conducted to elucidate the neurobiological substrates underlying these disorders. One biological mechanism being progressively investigated is stress-induced changes in the immune system that lead to neuroinflammation and alterations in the brain, with previous literature predominantly focusing on panic disorder, agoraphobia, and generalized anxiety disorder among the anxiety disorders. These disorders are associated with exaggerated fear reactions to stimuli specific to each disorder in the absence of any actual danger [3]. Panic disorder is characterized by recurrent unexpected panic attacks, with a panic attack being described as an abrupt surge of intense fear or discomfort that reaches a peak within minutes, during which time specific physical and cognitive symptoms occur. Individuals with agoraphobia report fear or anxiety in the following situations: using public transportation, being in open or enclosed places, standing in a line or being in a crowd, or being outside the home alone. Generalized anxiety disorder is marked by the presence of excessive anxiety and worry for at least 6 months, the worry is clearly excessive and experienced as very challenging to control, and are accompanied by physical or cognitive symptoms [2]. These disorders share underlying features associated with fear and anxiety, which indicates a common ground in neurobiological features, and are often found to be highly co-morbid with each other [4].

Immune system disruption has been implicated in anxiety disorders, with a majority of studies suggesting an increase in the inflammatory response. Conversely, no difference and a decrease in inflammatory marker levels have also been reported [5]. Stress has repeatedly been associated with anxiety disorders, as well as with the immune system; stressful life events that signal danger or threat have been specifically associated with the onset of anxiety disorders [6], and increases in various inflammatory markers have been reported in persons experiencing stressful conditions. Stress influences the hypothalamic–pituitary–adrenal axis and autonomic nervous system and ultimately the immune system, as chronic stress has been reported to lead to hypothalamic–pituitary–adrenal axis negative feedback disruption [7], enhanced sympathetic nervous system activity, and reduced parasympathetic nervous system activity [8], which in turn may increase proinflammatory cytokine levels [9]. Pro-inflammatory conditions can exert neurotoxic effects on the brain, with recent studies suggesting that anxiety symptoms represent brain alterations caused by neuro-inflammation [10]. Recent neuroimaging techniques are detecting objective biological markers that reflect the pathophysiology of various psychiatric disorders [11]. Identifying neural biomarkers of anxiety disorders, such as inflammation-associated changes in the brain, will further contribute to the early diagnosis of anxiety disorders [10]. In this review, we have focused on the influence of chronic stress on the hypothalamic-pituitary-adrenal axis, autonomic nervous system, and immune system, which leads to neuroinflammation and changes in brain structure and function that may be the underlying pathophysiology of anxiety disorders (i.e., panic disorder, agoraphobia and generalized anxiety disorder).

## 2. Chronic Stress, the Hypothalamic-Pituitary-Adrenal Axis, and the Autonomic Nervous System

Anxiety disorders have frequently been associated with chronic stress [12], with chronically stressful situations reported to precipitate the onset of panic disorder [13], agoraphobia [14], and generalized anxiety disorder [6]. One of the main stress response pathways is the hypothalamic-pituitary-adrenal axis [15]. As a reaction to stress, the hypothalamus releases corticotropin-releasing hormone that stimulates the anterior pituitary gland, which then secretes adrenocorticotropic hormone, and induces cortisol release from the adrenal glands. Hypothalamic-pituitary-adrenal axis negative feedback is controlled by cortisol binding to the anterior pituitary gland and hypothalamus. Under normal conditions, cortisol binds to the glucocorticoid receptor and acts as an anti-inflammatory agent [16] by inhibiting lymphocyte proliferation and decreasing the secretion of proinflammatory cytokines such as interleukin 6, interleukin 12, interferon gamma, and tumor necrosis factor alpha [17]. Glucocorticoid receptor binding also exerts anti-inflammatory effects by inhibiting nuclear factor kappa B [18]. However, chronic exposure to stress leads to excessive hypothalamic–pituitary–adrenal axis stimulation and hypercortisolemia [19]. Such excess secretion of cortisol may result in glucocorticoid receptor compensatory down-regulation or resistance, which hinders cortisol binding [20]. Excess cortisol increases mineralocorticoid receptor affinity, and when bound to the mineralocorticoid receptor, cortisol has proinflammatory effects [21]. Enhanced levels of inflammatory by-products can cause glucocorticoid receptor damage, which in turn leads to further cortisol pathway dysfunction [22]. Hypothalamic-pituitary-adrenal axis negative feedback is also disrupted by the impairment of glucocorticoid receptor binding, and cortisol levels which are normally sufficient are no longer able to inhibit corticotropin-releasing hormone release [23]. This in turn can activate inflammatory mast cells and stimulate norepinephrine release from the locus coeruleus [24]. As cortisol binding to the glucocorticoid receptor attenuates sympathetic nervous system activity [25], reduced levels of glucocorticoids can lead to increased sympathetic nervous system activity. Such dysfunctional activation of the hypothalamic–pituitary–adrenal axis, accompanied by changes in cortisol stress reactivity, have been reported in panic disorder, agoraphobia, and generalized anxiety disorder [26,27,28,29,30,31]. Schreiber et al. reported panic disorder patients had higher corticotropin-releasing hormone-induced adrenocorticotropic hormone and cortisol levels after the dexamethasone suppression/corticotropin-releasing hormone stimulation test [27]. Erhardt et al. reported hyper-responsiveness of the hypothalamic-pituitary-adrenal system in panic disorder patients after the dexamethasone suppression/corticotropin-releasing hormone stimulation test [28]. Elevated hair and salivary cortisol levels have been reported in individuals with generalized anxiety disorder [32] and panic disorder [33]. Increased cortisol awaking response and cortisol non-suppression in response to dexamethasone have been reported in patients with agoraphobia and panic disorder [34,35]. The results of these studies suggest the possible influence of chronic stress on hypothalamic–pituitary–adrenal axis activity and cortisol binding to the glucocorticoid receptor, which may contribute to the pro-inflammatory conditions observed in panic disorder, agoraphobia, and generalized anxiety disorder.

When a harmful event or threat is perceived, the sympathetic nervous system stimulates the body’s fight, flight, or freeze response, which are normal physiological reactions [36]. The amygdala initiates the reaction by triggering a neural response in the hypothalamus, which is followed by subsequent corticotropin-releasing hormone secretion that stimulates brainstem noradrenergic centers [37]. Sympathetic activity is activated through α1 adrenoceptors and parasympathetic activity is reduced through α2 adrenoceptors by the locus coeruleus in the brainstem [38,39]. Sympathetic nervous system activation then stimulates the adrenal medulla, which releases epinephrine and norepinephrine into the circulation. In normal conditions, the sympathetic nervous system is rapidly attenuated by parasympathetic nervous system activation, with acetylcholine subsequently being released when the stressor is terminated [40]. However, when stress exposure is prolonged, the sympathetic nervous system is continuously activated without the counteraction of the parasympathetic nervous system. As a result, catecholamine levels increase, and acetylcholine levels decrease [41]. Catecholamines then increase pro-inflammatory cytokines, as cytokine release is modulated by epinephrine and norepinephrine through α- and β-adrenoceptors expressed by immune cells [42]. On the other hand, acetylcholine decreases pro-inflammatory cytokines [43]. When stressful situations are chronic, sympathetic nervous system activation is continuous in the absence of parasympathetic nervous system counteraction, which induces an overall increase in catecholamine levels and a decrease in acetylcholine levels, which may ultimately lead to an increase in pro-inflammatory cytokine levels. Dysregulations of the autonomic nervous system and catecholamine activity have been suggested to be associated with the pathogenesis of anxiety disorders including panic disorder, agoraphobia, and generalized anxiety disorder [38]. Increased sympathetic tone, decreased parasympathetic tone, and compromised vagal tone have all been described in panic disorder [44]. With heart rate variability being considered as an index of the influence of both the sympathetic nervous system and the parasympathetic nervous system [45], high heart rate variability has been reported to reflect increased parasympathetic activity, and low heart rate variability to reflect increased sympathetic activity [46]. Decreased heart rate variability has been observed in patients with generalized anxiety disorder [47], along with a recent meta-analysis concluding anxiety disorders are associated with decreased heart rate variability [48]. The results of these studies suggest the possible influence of chronic stress on autonomic nervous system activity and levels of catecholamines and acetylcholine, which may contribute to the pro-inflammatory conditions observed in panic disorder, agoraphobia, and generalized anxiety disorder.

## 3. The Immune System in Anxiety Disorders

Although findings on inflammation associated with anxiety disorders have not been consistent, which to a certain extent is considered to be due to the wide range of variability among measures and samples, the overall evidence points to anxiety disorders being associated with enhanced inflammation [5]. C-reactive protein levels, which are increased following interleukin 6 secretion [49], were observed to be elevated in a large cohort of individuals diagnosed with panic disorder, agoraphobia, and generalized anxiety disorder [50]. Further increases in c-reactive protein levels were observed in generalized anxiety disorder patients [51,52]. Decreased concentrations of interleukin 2 were observed [53], with interleukin 2 amplifying CD8+ T cell responses or inducing regulatory T cell expansion, hence favoring either immune stimulation or suppression [54]. Decreased concentrations of interleukin 4, which have been reported to have anti-inflammatory properties [55], were also observed [53]. Increased tumor necrosis factor alpha levels have also been reported in generalized anxiety disorder patients [53], and genetic studies have reported patients with generalized anxiety disorder show alterations in immune-related gene expression [56]. De Berardis et al. investigated the relationships between alexithymia, suicide ideation, c-reactive protein, and serum lipid levels in patients with generalized anxiety disorder, and reported alexithymic generalized anxiety disorder patients show altered serum c-reactive protein levels [57]. Significant increases in c-reactive protein and tumor necrosis factor alpha levels were also observed in patients with agoraphobia over time [58]. Various studies have reported pro-inflammatory cytokines, including interleukin 1β, interleukin 6, and tumor necrosis factor alpha, are enhanced in panic disorder patients [59]. Studies on immune function have also reported circulating lymphocyte profile alterations and decreased cell activation in individuals with panic disorder [60,61]. Overall, preliminary evidence suggests anxiety disorders, including generalized anxiety disorder, agoraphobia and panic disorder, are associated with increased inflammation.

## 4. Systemic Inflammation and Neuroinflammation

Major pathways by which systemic inflammation can lead to neuroinflammation have been suggested. These involve neural routes, circumventricular organs, cytokine transport across the blood–brain barrier, and cytokine secretion by blood–brain barrier cells [62]. Early studies suggested that neural routes which communicate immune information to the brain exist, and that signals traveling through this route directly activate specific targets in the brain, without interference by the blood–brain barrier [63]. It was then first demonstrated that direct neural transmission via the vagus nerve was a critical neuroimmune communication pathway [64], and evidence supporting the afferent vagal pathway significantly accumulated in later studies. However, subsequent studies suggested that neuroimmune communication was not mediated by a single dominant pathway [65], and the concept of neural involvement could be expanded beyond that of vagal afferents [66]. It was then proposed that the inflammatory status of the entire body was monitored through numerous afferent pathways, and that the nervous system sensed immune activities throughout the body [67,68]. Circumventricular organs are regions of the brain in which the capillary bed does not form a blood–brain barrier [69]. As the vessels are leaky, circulating substances can reach these areas which communicate with other brain regions. Vice versa, substances produced within the brain can also be secreted into the blood through circumventricular organs [70]. An early study reported interleukin 1 was taken up by circumventricular organs from the systemic circulation [71], and other studies proposed that interleukin 1 works at its receptors and stimulates neural elements present in circumventricular organs, leading to signals being relayed to other brain areas [72,73]. A first study of cytokine transport across the blood–brain barrier reported a saturable transport system for cytokines and explained how molecules the size of cytokines could cross the blood–brain barrier [74]. The number of cytokines examined for blood–brain barrier transport expanded greatly thereafter, and further studies also reported that cytokine transporters were not static, but adapted to or were affected by physiological and disease states [75]. Regarding cytokine secretion by blood–brain barrier cells, brain endothelial cells were first reported to secrete cytokines such as interleukin 1 and interleukin 6 [76]. Thereafter, numerous cytokines have been found to be secreted by blood–brain barrier cells, including interleukin 3, interleukin 8, interleukin 10, endothelin 1, granulocyte macrophage colony-stimulating factor, monocyte chemoattractant protein 1, monokine induced by interferon-gamma, nerve growth factor, transforming growth factor beta, and tumor necrosis factor [62]. There is considerable evidence suggesting that systemic inflammation triggers a neuroinflammatory response, characterized by sustained microglial activation [77], with microglia being the primary source of cytokines in the inflamed central nervous system. Functionally, microglial activation is defined as the release of pro-inflammatory cytokines such as interleukin 1β, tumor necrosis factor α, and interleukin 6 [78]. Microglial cells also produce chemokines that draw monocytes to the brain [79], and monocytes in turn produce pro-inflammatory cytokines [80]. Such enhanced levels of pro-inflammatory cytokines in the brain may exert direct and indirect neurotoxic effects. The association between increased inflammation and anxiety disorders may be explained by neuroinflammation-induced toxic effects on specific brain regions implicated in each anxiety disorder.

## 5. Neurotoxic Cytokine Effects on the Brain

When brain cytokine networks are activated, certain brain areas are directly influenced through various mechanisms. The brain-derived neurotrophic factor signaling pathway was reported to be down-regulated by increased pro-inflammatory cytokines, which in turn leads to decreased neurotrophic support and neurogenesis [81,82,83]. Cell proliferation can be decreased through the nuclear factor kappa B signaling pathway by pro-inflammatory cytokines, as nuclear factor kappa B is the primary transcription factor for inflammatory response initiation, and conveys peripheral inflammatory signals to the central nervous system [84,85]. Glutamate levels can be increased by pro-inflammatory cytokines, inducing excitotoxicity and neurogenesis impairment through *N*-methyl-d-aspartate receptor activation [86]. Astrocytes and microglia can release reactive oxygen and nitrogen species when activated by pro-inflammatory cytokines, causing oxidative damage to neurons [87,88].

In the kynurenine pathway, kynurenine is formed when tryptophan 2,3-dioxygenase cleaves the indole-ring of tryptophan [89]. Indoleamine 2,3-dioxygenase also initiates the kynurenine pathway, although its activity is minimal under normal conditions [36]. Kynurenine is then metabolized to kynurenic acid through kynurenine amino-transferase, anthranilic acid through kynureninase, and 3-hydroxykynurenine through kynurenine monooxygenase. The 3-hydroxykynurenine is converted into 3-hydroxyanthranilic acid, and eventually quinolinic acid [90,91]. Peripheral kynurenine is a brain penetrant and initiates kynurenine metabolism in the brain [92,93]. Different arms of the kynurenine metabolism take place in different brain cell types. The 3-hydroxykynurenine metabolism that results in quinolinic acid production takes place in the microglia via kynurenine monooxygenase and kynurenic acid production taking place in astrocytes through kynurenine amino-transferase [94]. Pro-inflammatory cytokines increase indoleamine 2,3-dioxygenase activity [95,96,97], and as a result more tryptophan degradation occurs through the kynurenine pathway, which increases the kynurenine/tryptophan ratio [98]. As pro-inflammatory cytokines induce increases in brain penetrant kynurenine and consequent kynurenine monooxygenase activity, downstream kynurenine metabolites also increase [96]. Kynurenine metabolites such as 3- hydroxykynurenine, 3-hydroxyanthranilic acid, and quinolinic acid are all neurotoxic [89]. On the other hand, kynurenic acid, which is reduced in inflammatory states, counteracts neurotoxicity [62]. Such ongoing imbalances between neurotoxic and neuroprotective kynurenine metabolites can lead to brain alterations.

Numerous studies have reported increased cytokine levels influence prefrontal and limbic structures [99], which are areas that have repeatedly been associated with anxiety disorders [100]. Increases in interleukin 6 and tumor necrosis factor alpha were associated with laboratory-based stressors in healthy controls, with tumor necrosis factor alpha levels showing positive correlations with anterior cingulate and insula activities [101]. Increased levels of tumor necrosis factor alpha, interleukin 1, and interleukin 6 were reported after bacterial lipopolysaccharide injections, accompanied by an increased state of anxiety and orbitofrontal cortex activation in response to emotional visual stimuli [102]. Enhanced pro-inflammatory cytokine levels following low-dose endotoxin injections were shown to decrease ventral striatum activity for an anticipated reward, with decreased limbic system activity being associated with the ventral striatum [103]. An enhanced interleukin 6 level induced by typhoid vaccination was accompanied by mood deterioration, which showed positive correlations with subgenual anterior cingulate activity. Furthermore, decreased connectivity between the subgenual anterior cingulate and the nucleus accumbens, superior temporal sulcus, medial prefrontal cortex, and amygdala were observed. Levels of interleukin 6 were suggested to modulate subgenual anterior cingulate connectivity to these regions [104]. Increases in interleukin 6 levels showed positive associations with amygdala activity in reaction to laboratory-based stressors, with increased amygdala and dorsomedial prefrontal cortex coupling [105]. Enhanced interleukin 6 levels were accompanied by fatigue, confusion, and impaired concentration [106] and were reported to influence amygdala and subgenual anterior cingulate cortex connectivity [104]. Enhanced levels of interleukin 6 and tumor necrosis factor alpha were shown to increase amygdala reactivity in response to stress, accompanied by feelings of social disconnection [107]. Stronger coupling between the dorsomedial prefrontal cortex and amygdala was associated with an increased inflammatory response to a stressor [105]. Such preliminary evidence suggests neuroinflammation is associated with alterations in prefrontal and limbic structures.

## 6. Alterations of Limbic and Pre-Frontal Structures of the Brain in Anxiety Disorders

Previous imaging studies have reported brain structural and functional changes in anxiety disorders, with emphasis being made on limbic and pre-frontal structures in particular [108]. For panic disorder, changes in volumes of the frontal and orbitofrontal cortices [109,110,111,112,113,114], anterior cingulate cortex [109,112,115,116], amygdala [109,117,118,119,120], temporal lobe [110,111,120,121], parahippocampal gyrus [122], insula [109], basal ganglia [111,118], caudate [123,124], and brainstem [112,116,125] have been reported. Volume reductions were reported in the prefrontal cortex [109,112], frontal lobe [110], frontal gyrus [111], orbitofrontal cortex [110,113], anterior cingulate cortex [109,115,116,124], precuneus [111], caudate [123], amygdala [109,117,119,120], hippocampus [120], insular cortex [109], temporal gyrus [109,110,111], temporal lobe [110,120,121], parahippocampal gyrus [122,123], putamen [111], and midbrain [125]. Increased volumes were reported in the hippocampus [112], temporal gyrus, insula [116], midbrain, and pons [112,116]. Studies on white matter connectivity in panic disorder patients have reported increased fractional anisotropy values in cingulate regions [126]. Studies on resting state function in panic disorder patients have reported changes in the thalamus, along with frontal and temporal lobes [127,128,129,130,131]. Increased levels of glucose uptake were observed in the orbitofrontal cortex [131], amygdala [127], hippocampus [127,128], parahippocampal area [128], thalamus, midbrain, pons, medulla, cerebellum [127], and occipital cortex [130]. Decreased levels of glucose metabolism were observed in the anterior cingulate [131], hippocampal [130], parietal [128,131], and temporal [128,129] brain regions.

Structural changes in prefrontal areas such as the orbitofrontal cortex have been reported in patients with agoraphobia symptoms [132]. Patients with panic disorder with agoraphobia showed decreased gray matter volume in the left medial orbitofrontal gyrus, while patients with panic disorder without agoraphobia did not show any decrease in gray matter volume. Also, previous studies have suggested hippocampal activity and networks involving the prefrontal cortex and amygdala coupling play a role in agoraphobia symptoms [133,134,135]. Lueken et al. reported panic disorder patients with agoraphobia exhibited enhanced activation in the right pregenual anterior cingulate cortex, hippocampus, and amygdala in response to a safety signal [133]. Wittmann et al. reported activations in areas associated with the fear circuit including the amygdala, insula, and hippocampal areas in panic disorder patients with agoraphobia, in an fMRI paradigm with agoraphobia-specific stimuli [134]. Lueken et al. reported enhanced activation of the bilateral dorsal inferior frontal gyrus and the midbrain in panic disorder patients with agoraphobia during a fear conditioning task [135]. Hyperactivation of the ventral striatum and insula when anticipating agoraphobia-specific situations was also suggested as a central neurofunctional correlate of agoraphobia [136].

For generalized anxiety disorder, structural alterations of the amygdala [137], hippocampus [138], superior temporal gyrus [139], prefrontal cortex [140], anterior cingulate cortex [141], hypothalamus [142], thalamus [143], and basal ganglia [144] have been reported. Volume reductions were reported in the dorsolateral prefrontal cortex [144], hypothalamus [142], insula, hippocampus, thalamus, superior temporal gyrus, and midbrain [143]. Increased volumes were reported in the dorsomedial prefrontal cortex [140], amygdala [137,140], superior temporal gyrus [139], superior temporal pole, and basal ganglia [144]. Functional studies on generalized anxiety disorder patients have reported changes in activity of the amygdala [145,146], hippocampus [147], prefrontal cortex [146,148,149], frontal gyrus [150], anterior cingulate cortex [148,149,151], ventral tegmental area [152], thalamus [153], striatum, and insula [154]. Exaggerated responses to stimuli were reported in the prefrontal cortex [148,149,151,153], anterior cingulate cortex [148,149], frontal gyrus [150], amygdala [145,146,148,153], ventral tegmental area [152], and thalamus [153]. Reduced activities were reported in the ventromedial prefrontal cortex [153,154], anterior cingulate cortex [153], striatum, insula, and paralimbic regions [154]. Changes in connectivity between the amygdala, prefrontal cortex, and anterior cingulate cortex have also widely been reported in patients with generalized anxiety disorder [146,155,156,157]. Hilbert et al. reported abnormal amygdala and prefrontal cortex activation and decreased functional connectivity between these regions in generalized anxiety disorder patients [155]. Etkin et al. reported the connectivity patterns between basolateral subregions of the amygdala and medial prefrontal cortices, and connectivity patterns between centromedial subregions of the amygdala and the midbrain, thalamus, and cerebellum were significantly less robust in generalized anxiety disorder patients [156]. Roy et al. reported adolescents with generalized anxiety disorder exhibited disruptions in amygdala-based intrinsic functional connectivity networks that include regions in the medial prefrontal cortex, insula, and cerebellum [157]. Such preliminary evidence suggests alterations in limbic and prefrontal structures are associated with the pathophysiology of anxiety disorders (i.e., panic disorder, agoraphobia, and generalized anxiety disorder).

## 7. Conclusions

Chronic stress may lead to hypothalamic–pituitary–adrenal axis and autonomic nervous system disruption, which in turn may induce systemic proinflammatory conditions. Systemic inflammation leads to neuroinflammation, and enhanced levels of pro-inflammatory cytokines in the brain exert neurotoxic effects on specific brain regions, either directly or secondarily through the kynurenine pathway. This may cause alterations in the structure or function of anxiety-related brain circuits (mainly limbic and pre-frontal structures) priming the brain to be vulnerable to anxiety disorders (Figure 1). Further research on the use of inflammatory markers and brain imaging in the early diagnosis of panic disorder, agoraphobia, and generalized anxiety disorder are needed in order to better identify and manage these disorders.

As stress-induced inflammatory conditions have repeatedly been suggested to underlie the pathophysiology of various psychiatric disorders, previous studies have investigated the possible therapeutic role of anti-inflammatory agents in conditions such as depression [158]. Nonsteroidal anti-inflammatory drugs, cytokine-inhibitors, statins, poly-unsaturated fatty acids, pioglitazone, corticosteroids, minocycline, and modafinil have all been studied for their potential antidepressant treatment effects. However, the efficacy of anti-inflammatory treatment agents in anxiety disorders has not yet been widely investigated. Other behavioral interventions that have been shown to be effective in alleviating anxiety symptoms, such as yoga and meditation [159], have been suggested to do so by dampening inflammatory processes [160]. Although mixed effects were shown for the association between such interventions and circulating inflammatory markers [161], more consistent findings were seen for genomic markers, with trials showing decreased expression of inflammation-related genes and reduced signaling [162,163]. Furthermore, the importance of microbiota-gut brain axis dysregulation in stress-related disorders, possibly through modulating inflammatory pathways, has been investigated in recent studies [164]. Certain probiotics have been shown to improve anxiety symptoms and have even been termed as psychobiotics [165]. Adherence to Mediterranean dietary patterns has also been suggested to improve anxiety symptoms [166], with Mediterranean diets being reported to modulate inflammatory processes [167].

The continuous activation of the sympathetic nervous system without the counteraction of the parasympathetic nervous system leads to increases in catecholamine levels and decreases in acetylcholine levels [41], which in turn leads to increases in pro-inflammatory cytokine levels [43]. Therefore, pharmacological and non-pharmacological approaches that attenuate sympathetic nervous system activity and decrease catecholamine activity, activate the parasympathetic nervous system, and increase cholinergic activity have been suggested as possible treatment methods that suppress inflammation. As hypertension has been associated with several modifications in the function and regulation of the sympathetic nervous system, several antihypertensive medications are considered to exert an influence on sympathetic nervous system function, such as β-blockers, α-blockers, and centrally acting drugs [168]. Propranolol is a ß1,2-adrenoreceptor antagonist which competes at the receptor level with catecholamines, thereby blocking their orthosympathetic effects [169]. Only a few systematic reviews on the effects of propranolol for the treatment of anxiety disorders have been conducted, with a recent meta-analysis reporting the quality of evidence for the efficacy of propranolol to be insufficient to support its routine use for the treatment of anxiety disorders [170]. However, propranolol has been widely used off-label for anxiety [171], drug withdrawal symptoms [172], aggression [173], performance anxiety related to examinations [174], on stage [175], among musicians [176], among surgeons [177], for patients who fear undergoing surgery [178], and in the field of psychiatry. Recent animal studies have reported the effects of propranolol on neuroinflammation. Wohleb et al. reported stress-induced neuroinflammation, characterized by increased inflammatory markers such as interleukin-1β on the surface of microglia and macrophages, was prevented by propranolol [179]. Sugama et al. reported brain microglial activation due to acute stress occurred mostly in the hippocampus, thalamus, and hypothalamus, with the noradrenaline synthesizing enzyme (dopamine β-hydroxylase) being densely stained in the neuronal fibers of these brain regions, and propranolol treatment inhibited microglial activation in terms of morphology and count through the whole brain [180]. Armstead et al. reported propranolol reduced the upregulation of interleukin-6 and prevented neuronal cell death in the cornu amonis 1 and cornu amonis 3 of the hippocampus [181]. Lin et al. provided experimental evidence showing the suppressive effects of propranolol on inflammation and brain injury. Pretreatment with propranolol was shown to protect against postischemic brain infarction, edema, and apoptosis; the neuroprotection caused by propranolol was accompanied by a reduction in plasma c-reactive protein, plasma free fatty acids, plasma corticosterone, brain oxidative stress, and brain inflammation [182]. Further systematic research on the effects of propranolol treatment for anxiety disorders, and the possible mechanisms of such treatment effects, i.e., attenuation of neuroinflammation and consequent neurotoxic effects, should be conducted.

Vagus nerve stimulation has been reported to attenuate the systemic inflammatory response [67], which is mediated by the efferent vagus nerve, with acetylcholine being the neurotransmitter for pre- and postganglionic vagal efferent nerves [183]. The term “cholinergic anti-inflammatory pathway” was then coined, and subsequent studies established that acetylcholine and nicotine attenuated inflammation by stimulating alpha-7 nicotinic receptors [68]. Therefore, the clinical application of vagus nerve stimulation and alpha-7 nicotinic receptor agonists have been suggested in various medical fields. Consequently, cholinergic therapies are being investigated using animal models of sepsis [184], burn injury [185], rheumatoid arthritis [186], inflammatory bowel disease [187], stroke [188], and traumatic brain injury [189]. In the field of psychiatry, vagus nerve stimulation was approved by the Food and Drug Administration in 2005 for the treatment of patients with chronic or recurrent unipolar or bipolar depression who had a history of failing four antidepressant interventions [190]. Vagus nerve stimulation modulates neural circuitry by stimulating vagal afferent fibers in the neck, with the vagus nerve being connected to brain areas such as the locus coeruleus, amygdala, hippocampus, orbito-frontal cortex, and insular cortex, which are all involved in emotional and cognitive processing. Although vagus nerve stimulation is being suggested as a potential therapeutic modality for anxiety disorders, only a few systematic studies have been conducted. A pilot study reported the short and long-term efficacies of vagus nerve stimulation in patients with treatment-resistant obsessive-compulsive disorder, panic disorder, and posttraumatic stress disorder [191]. Less invasive or non-invasive alternatives to vagus nerve stimulation are also under investigation, such as transvenous vagus stimulation [192] or systems that provide transcutaneous vagus nerve stimulation [193]. Furthermore, pharmacological agents that activate cholinergic anti-inflammatory mechanisms are being studied, such as nicotine, selective alpha-7 nicotinic receptor agonists, or positive allosteric modulators of the alpha-7 nicotinic receptor [183]. Additional studies are needed in order to investigate the potential efficacies these pharmacological and non-pharmacological cholinergic therapies may have for the treatment of anxiety disorders (i.e., panic disorder, agoraphobia and generalized anxiety disorder).

Further systematic research on the use of inflammatory markers and brain imaging in the early diagnosis of panic disorder, agoraphobia, and generalized anxiety disorder, and the therapeutic efficacy of anti-inflammatory interventions, will help to develop effective ways to detect and treat anxiety disorders.

## Figures and Tables

**Figure 1 ijms-21-06546-f001:**
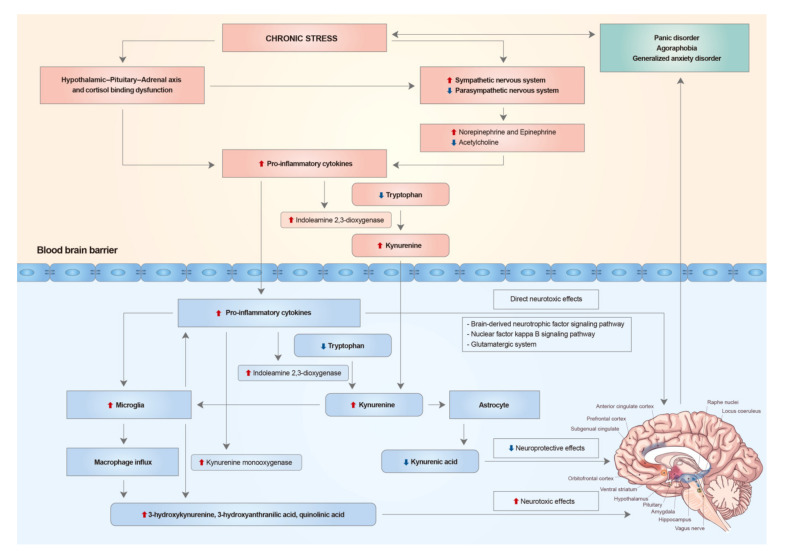
Chronic stress may lead to hypothalamic–pituitary–adrenal axis and autonomic nervous system disruption, which in turn may induce systemic proinflammatory conditions. Systemic inflammation leads to neuroinflammation, and enhanced levels of pro-inflammatory cytokines in the brain exert neurotoxic effects on specific brain regions, either directly or secondarily through the kynurenine pathway. This may cause alterations in the structure or function of anxiety-related brain circuits, mainly limbic and pre-frontal structures, priming the brain to be vulnerable to anxiety disorders, such as panic disorder, agoraphobia and generalized anxiety disorder.

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
