# Peer review of "Neuroinflammation-Associated Alterations of the Brain as Potential Neural Biomarkers in Anxiety Disorders"

_ijms, 2020, doi:10.3390/ijms21186546_

Round 1

Reviewer 1 Report

This is well-written manuscript that provides very broad coverage of relevant topics. However, it tends to err on the side of being too broad and rather superficial in its coverage. There is not a lot of critical thinking evident and most of this work has been well covered elsewhere.  The paper lacked a clear statement of its goals and review methodology.  I found the most interesting part of it to be sections that addressed the relationship between peripheral inflammation and central nervous system inflammation and I think it could make a stronger contribution to the literature if it honed-in on mechanisms of blood-brain barrier permeability and transport.  The  nice figure emphasizes the relationship between peripheral and central markers and readers may be more interested in a more focused analysis of that component of the model.

Author Response

IJMS-883186

We greatly appreciate your editorial decision that has allowed us to improve our manuscript. We have revised the manuscript in accordance with the Reviewers’ comments and suggestions. We respond here in detail to each of the Reviewer’s comments:

 Reviewer 1

This is well-written manuscript that provides very broad coverage of relevant topics. However, it tends to err on the side of being too broad and rather superficial in its coverage. There is not a lot of critical thinking evident and most of this work has been well covered elsewhere. The paper lacked a clear statement of its goals and review methodology. I found the most interesting part of it to be sections that addressed the relationship between peripheral inflammation and central nervous system inflammation and I think it could make a stronger contribution to the literature if it honed-in on mechanisms of blood-brain barrier permeability and transport. The nice figure emphasizes the relationship between peripheral and central markers and readers may be more interested in a more focused analysis of that component of the model.

  • We appreciate the Reviewer's positive evaluation of our manuscript. We thank the Reviewer for the thorough evaluation of the manuscript as well as for the critical and helpful comments.
  • We agree with the Reviewer, and have elaborated on the second paragraph of Section 3 (The immune system in anxiety disorders), of the original manuscript. The following sentences of the original manuscript “Systemic inflammation can lead to neuroinflammation as pro-inflammatory cytokines can cross the blood–brain barrier [60, 61]. Cytokine receptor activation can also convey pro-inflammatory cytokine signals to the brain [62]. Pro-inflammatory cytokines, nitric oxide, and reactive oxygen species are produced by microglia, which are the primary inflammatory neuronal cells [63]. Microglial cells also produce chemokines that draw monocytes to the brain [64], and monocytes in turn produce pro-inflammatory cytokines [65]. Enhanced levels of pro-inflammatory cytokines may exert direct and indirect neurotoxic effects. The association between increased inflammation and anxiety disorders may be explained by neuroinflammation-induced toxic effects on specific brain regions implicated in each anxiety disorder” have been modified and included in a separate section titled “4. Systemic inflammation and neuroinflammation”.

 Lines 169-209: 4. Systemic inflammation and neuroinflammation

Major pathways by which systemic inflammation can lead to neuroinflammation have been suggested. These involve neural routes, circumventricular organs, cytokine transport across the blood–brain barrier, and cytokine secretion by blood–brain barrier cells [62]. Early studies suggested that neural routes which communicate immune information to the brain exist, and that signals traveling through this route directly activate specific targets in the brain, without interference by the blood–brain barrier [63]. It was then first demonstrated that direct neural transmission via the vagus nerve was a critical neuroimmune communication pathway [64], and evidence supporting the afferent vagal pathway has significantly accumulated in later studies. However, subsequent studies suggested that neuroimmune communication was not mediated by a single dominant pathway [65], and the concept of neural involvement could be expanded beyond that of vagal afferents [66]. It was then proposed that the inflammatory status of the entire body was monitored through numerous afferent pathways, and that the nervous system sensed immune activities throughout the body [67, 68]. Circumventricular organs are regions of the brain in which the capillary bed does not form a blood–brain barrier [69]. As the vessels are leaky, circulating substances can reach these areas which communicate with other brain regions. Vice versa, substances produced within the brain can also be secreted into the blood through circumventricular organs [70]. An early study reported interleukin 1 was taken up by circumventricular organs from the systemic circulation [71], and other studies proposed that interleukin 1 works at its receptors and stimulates neural elements present in circumventricular organs, leading to signals being relayed to other brain areas [72, 73]. A first study of cytokine transport across the blood–brain barrier reported a saturable transport system for cytokines and explained how molecules the size of cytokines could cross the blood–brain barrier [74]. The number of cytokines examined for blood–brain barrier transport expanded greatly thereafter, and further studies also reported that cytokine transporters were not static, but adapted to or were affected by physiological and disease states [75]. Regarding cytokine secretion by blood–brain barrier cells, brain endothelial cells were first reported to secrete cytokines such as interleukin 1 and interleukin 6 [76]. Thereafter, numerous cytokines have been found to be secreted by blood–brain barrier cells, including interleukin 3, interleukin 8, interleukin 10, endothelin 1, granulocyte macrophage colony- stimulating factor, monocyte chemoattractant protein 1, monokine induced by interferon-gamma, nerve growth factor, transforming growth factor beta, and tumor necrosis factor [62]. There is considerable evidence suggesting that systemic inflammation triggers a neuroinflammatory response, characterized by sustained microglial activation [77], with microglia being the primary source of cytokines in the inflamed central nervous system. Functionally, microglial activation is defined as the release of pro‐inflammatory cytokines such as interleukin 1β, tumor necrosis factor α, and interleukin 6 [78]. Microglial cells also produce chemokines that draw monocytes to the brain [79], and monocytes in turn produce pro-inflammatory cytokines [80]. Such enhanced levels of pro-inflammatory cytokines in the brain may exert direct and indirect neurotoxic effects. The association between increased inflammation and anxiety disorders may be explained by neuroinflammation-induced toxic effects on specific brain regions implicated in each anxiety disorder.

Reviewer 2 Report

The review paper from Won and Kim entitled “Neuroinflammation-associated alterations of the brain as potential neural biomarkers in anxiety disorders” presents pieces of evidence, extrapolated from the literature, around the interesting connection of anxiety disorders with inflammation. It’s my opinion that the manuscript may be published provided that extensive but merely formal corrections are done. Indeed, especially in the 5th paragraph, which accounts for a great part of the manuscript, a more narrative and less "listing" description of the current and past literature would greatly benefit the readership by avoiding some boredom. Also, the elliptical avoidance of the word "that" is in my opinion excessively employed throughout the text.

Specific points:

Line 57: ...as well as WITH the immune...

Line 153: IL-2 is a type 1 cytokine involved in proliferation of a plethora of lymphoid cells also with pro-inflammatory potential, thus its definition by the authors as anti-inflammatory is at odd with current knowledge. Instead, the cited study of ref 53 suggest a Th17-skewed immune profile in GAD patients, as opposed to Th1 and Th2, and not a mere anti- vs pro-inflammatory response

Line 170. Microglial cells are not "neuronal" (they originates from yolk sac derived hematopoietic precursors, thus not sharing with neurons the ectodermal origin), nor, in principle, inflammatory. Instead, a definition fitting with the text should be "the primary source of cytokines in the inflamed CNS"

The fifth paragraph suffers from poor narrative shape. From row 237 to 265 and so on, a very long list of studies is rattled off in the form of: Author A et al reported...Author B et al., reported...etc. This may result in annoying the reader. A more narrative writing should be done, perhaps at the cost of reducing the number of cited papers.

Author Response

IJMS-883186

We greatly appreciate your editorial decision that has allowed us to improve our manuscript. We have revised the manuscript in accordance with the Reviewers’ comments and suggestions. We respond here in detail to each of the Reviewer’s comments:

Reviewer 2

The review paper from Won and Kim entitled “Neuroinflammation-associated alterations of the brain as potential neural biomarkers in anxiety disorders” presents pieces of evidence, extrapolated from the literature, around the interesting connection of anxiety disorders with inflammation. It’s my opinion that the manuscript may be published provided that extensive but merely formal corrections are done. Indeed, especially in the 5th paragraph, which accounts for a great part of the manuscript, a more narrative and less "listing" description of the current and past literature would greatly benefit the readership by avoiding some boredom. Also, the elliptical avoidance of the word "that" is in my opinion excessively employed throughout the text.

  • We appreciate the Reviewer's positive evaluation of our manuscript. We thank the Reviewer for the thorough evaluation of the manuscript as well as for the critical and helpful comments. We respond here in detail to each of the Reviewer’s comments.

 Line 57: ...as well as WITH the immune...

  • Thank you for the comment. We have made the following change in lines 56-57 of the original manuscript (Stress has repeatedly been associated with anxiety disorders as well as the immune system)

Lines 57-58: Stress has repeatedly been associated with anxiety disorders as well as with the immune system

 Line 153: IL-2 is a type 1 cytokine involved in proliferation of a plethora of lymphoid cells also with pro-inflammatory potential, thus its definition by the authors as anti-inflammatory is at odd with current knowledge. Instead, the cited study of ref 53 suggest a Th17-skewed immune profile in GAD patients, as opposed to Th1 and Th2, and not a mere anti- vs pro-inflammatory response

  • Thank you for the comment. We agree with the Reviewer and have made the following changes in lines 152-153 (Further increases in c-reactive protein levels were observed in generalized anxiety disorder patients [51, 52], along with decreased concentrations of anti-inflammatory cytokines such as interleukin 2 and interleukin 4 [53]) of the original manuscript, as follows.

Lines 151-156: Further increases in c-reactive protein levels were observed in generalized anxiety disorder patients [51, 52]. Decreased concentrations of interleukin 2 were observed [53], with interleukin 2 amplifying CD8+ T cell responses or inducing regulatory T cell expansion, hence favoring either immune stimulation or suppression [54]. Decreased concentrations of interleukin 4, which have been reported to have anti-inflammatory properties [55], were also observed [53].

Line 170. Microglial cells are not "neuronal" (they originates from yolk sac derived hematopoietic precursors, thus not sharing with neurons the ectodermal origin), nor, in principle, inflammatory. Instead, a definition fitting with the text should be "the primary source of cytokines in the inflamed CNS"

  • Thank you for the comment. We agree with the Reviewer and have modified lines 169-170 (Pro-inflammatory cytokines, nitric oxide, and reactive oxygen species are produced by microglia, which are the primary inflammatory neuronal cells [63]) of the original manuscript as follows.

Lines 199-204: There is considerable evidence suggesting that systemic inflammation triggers a neuroinflammatory response, characterized by sustained microglial activation [77], with microglia being the primary source of cytokines in the inflamed central nervous system. Functionally, microglial activation is defined by the release of pro‐inflammatory cytokines such as interleukin 1β, tumor necrosis factor α, and interleukin 6 [78].

 The fifth paragraph suffers from poor narrative shape. From row 237 to 265 and so on, a very long list of studies is rattled off in the form of: Author A et al reported...Author B et al., reported...etc. This may result in annoying the reader. A more narrative writing should be done, perhaps at the cost of reducing the number of cited papers.

  • Thank you for the comment. We agree with the Reviewer and have made the following changes in Section 6 (Alterations of limbic and pre-frontal structures of the brain in anxiety disorders) of the original manuscript.

Lines 237-265 of the original manuscript: Asami et al. reported gray matter volume reductions in the bilateral dorsomedial and right ventromedial prefrontal cortices, right amygdala, anterior cingulate cortex, bilateral insular cortex, occipitotemporal gyrus, and left cerebellar vermis in panic disorder patients [94]. Sobanski et al. reported volume reductions of bilateral temporal lobes, right frontal lobe, right middle temporal gyrus, and medial orbitofrontal cortex in patients with panic disorder [95]. Yoo et al. reported decreased volumes of bilateral putamen, right precuneus, right inferior temporal gyrus, right inferior frontal gyrus, left superior temporal gyrus, and left superior frontal gyrus in panic disorder patients [96]. Protopopescu et al. reported increased volumes of the midbrain, rostral pons of the brainstem, and ventral hippocampus, and decreased volume of the prefrontal cortex in patients with panic disorder [97]. Roppongi et al. reported volume reductions in the right posterior-medial orbitofrontal cortex in panic disorder patients [98]. Lai et al. reported grey-matter deficits in infero-frontal, limbic, occipital, temporo-parietal, and cerebellar areas in patients with major depressive disorder combined with panic disorder [99]. Asami et al. reported volume reduction in the right dorsal anterior cingulate cortex in panic disorder patients [100]. Uchida et al. reported volume increases in the left insula, left superior temporal gyrus, midbrain, and pons, and volume decreases in the right anterior cingulate cortex in panic disorder patients [101]. Hayano et al. reported volume reductions in bilateral amygdala of patients with panic disorder [102]. Lai et al. reported decreased volumes of the right anterior cingulate cortex, right medial frontal gyrus, left posterior cingulate cortex, right parahippocampal gyrus, limbic areas, occipital lingual gyrus, and bilateral cerebellum in patients with major depressive disorder and panic disorder [103]. Massana et al. reported volume reduction in bilateral amygdala in panic disorder patients [104]. Uchida et al. reported decreased volumes of the bilateral temporal lobe, bilateral amygdala, and left hippocampus in patients with panic disorder [105]. Vythilingam et al. reported smaller volumes of the bilateral temporal lobes in panic disorder patients [106]. Massana et al. reported lower density of the left parahippocampal gyrus in patients with panic disorder [107]. Lai et al. reported decreased volumes in the right caudate head and right parahippocampal gyrus in panic disorder patients [108]. Radua et al. reported patients with anxiety disorders including panic disorder showed decreased volumes of the bilateral dorsomedial frontal/ anterior cingulate gyri [109]. Fujiwara et al. reported increased dorsal midbrain volume in panic disorder patients [110].

Lines 271-276: Volume reductions were reported in the prefrontal cortex [109, 112], frontal lobe [110], frontal gyrus [111], orbitofrontal cortex [110, 113], anterior cingulate cortex [109, 115, 116, 124], precuneus [111], caudate [123], amygdala [109, 117, 119, 120], hippocampus [120], insular cortex [109], temporal gyrus [109-111], temporal lobe [110, 120, 121], parahippocampal gyrus [122, 123], putamen [111], and midbrain [125]. Increased volumes were reported in the hippocampus [112], temporal gyrus, insula [116], midbrain, and pons [112, 116].

 Lines 268-279 of the original manuscript: Sakai et al. reported panic disorder patients exhibited significantly higher levels of glucose uptake in the bilateral amygdala, hippocampus, and thalamus, and in the midbrain, caudal pons, medulla, and cerebellum [112]. Bisaga et al. reported significantly increased glucose metabolism in the left hippocampus and parahippocampal area and significantly decreased glucose metabolism in the right inferior parietal and right superior temporal brain regions in patients with panic disorder [113]. Lee et al. reported decreased regional cerebral blood flow in the right superior temporal lobe in patients with panic disorder [114]. De Cristofaro et al. reported increased blood flow in the left occipital cortex and decreased blood flow in the bilateral hippocampal regions in panic disorder patients [115]. Nordahl et al. reported metabolic decreases in the left inferior parietal lobule and anterior cingulate, and metabolic increases in the medial orbital frontal cortex of panic disorder patients [116].

Lines 280-284: Increased levels of glucose uptake were observed in the orbitofrontal cortex [131], amygdala [127], hippocampus [127, 128], parahippocampal area [128], thalamus, midbrain, pons, medulla, cerebellum [127], and occipital cortex [130]. Decreased levels of glucose metabolism were observed in the anterior cingulate [131], hippocampal [130], parietal [128, 131], and temporal [128, 129] brain regions.

 Lines 296-310 of the original manuscript: De Bellis et al. reported amygdala volumes to be larger in generalized anxiety disorder patients [122]. Cha et al. reported patients with only generalized anxiety disorder, and patients with comorbid generalized anxiety disorder and major depressive disorder showed abnormal microstructure in the cornu amonis 1 and cornu amonis 2-3 of the hippocampus [123]. De Bellis et al. reported pediatric generalized anxiety disorder patients showed increased superior temporal gyrus volumes [124]. Schienle et al. reported patients with generalized anxiety disorder had larger volumes of the amygdala and the dorsomedial prefrontal cortex [125]. Zhang et al. reported generalized anxiety disorder patients showed higher fractional anisotropy in the right amygdala white matter and lower fractional anisotropy in the caudal anterior cingulate cortex/ mid-cingulate cortex white matter [126]. Terlevic et al. reported hypothalamus volume was reduced in generalized anxiety disorder patients [127]. Moon et al. reported patients with generalized anxiety disorder showed volume reductions in the hippocampus, midbrain, thalamus, insula, and superior temporal gyrus [128]. Hilbert et al. reported higher volumes of the basal ganglia and superior temporal pole, and lower volumes of the dorsolateral prefrontal cortex white matter in patients with generalized anxiety disorder [129].

Lines 302-306: Volume reductions were reported in the dorsolateral prefrontal cortex [144], hypothalamus [142], insula, hippocampus, thalamus, superior temporal gyrus, and midbrain [143]. Increased volumes were reported in the dorsomedial prefrontal cortex [140], amygdala [137, 140], superior temporal gyrus [139], superior temporal pole, and basal ganglia [144].

 Lines 313-333 of the original manuscript: Thomas et al. reported pediatric patients with generalized anxiety disorder or panic disorder showed exaggerated amygdala response to fearful faces [130]. Monk et al. reported generalized anxiety disorder patients to show greater right amygdala activation when viewing masked angry faces [131]. Chen et al. reported decreased anterior hippocampal connectivity in generalized anxiety disorder patients [132]. McClure et al. reported generalized anxiety disorder patients showed greater activation to fearful faces than to happy faces in a distributed network including the amygdala, ventral prefrontal cortex, and anterior cingulate cortex [133]. Paulesu et al. reported generalized anxiety disorder subjects showed persistent activation of the anterior cingulate and dorsal medial prefrontal cortex even during resting state scans [134]. Blair Paulesu et al. reported patients with generalized anxiety disorder showed increased middle frontal gyrus responses to angry expressions [135]. Krain et al. reported intolerance of uncertainty was positively correlated with activity in several frontal and limbic regions in adolescent patients with generalized anxiety disorder and/or social phobia [136]. Cha et al. reported patients with generalized anxiety disorder showed heightened and less discriminating ventral tegmental area reactivity to generalization stimuli [137]. Buff et al. reported anxiety inducing and arousing disorder-related scripts to elicit elevated activity in the amygdala, dorsomedial prefrontal cortex, ventrolateral prefrontal cortex, and the thalamus, as well as reduced activity in the ventromedial prefrontal cortex and subgenual anterior cingulate cortex in generalized anxiety disorder patients [138]. Hoehn-Saric et al. reported worry sentences elicited reduced blood oxygen level dependent responses in prefrontal regions, the striatum, insula, and paralimbic regions in generalized anxiety disorder patients [139].

Lines 309-313: Exaggerated responses to stimuli were reported in the prefrontal cortex [148, 149, 151, 153], anterior cingulate cortex [148, 149], frontal gyrus [150], amygdala [145, 146, 148, 153], ventral tegmental area [152], and thalamus [153]. Reduced activities were reported in the ventromedial prefrontal cortex [153, 154], anterior cingulate cortex [153], striatum, insula, and paralimbic regions [154].

Round 2

Reviewer 2 Report

The authors made all amendments suggested by the reviewer